# Limitations of Recent Studies Dealing with the Antibacterial Properties of Silver Nanoparticles: Fact and Opinion

**DOI:** 10.3390/nano9121775

**Published:** 2019-12-13

**Authors:** Raphaël E. Duval, Jimmy Gouyau, Emmanuel Lamouroux

**Affiliations:** 1Université de Lorraine, CNRS, L2CM, F-54000 Nancy, France; jimmy.gouyau@univ-lorraine.fr; 2ABC Platform^®^, Faculté de Pharmacie, F-54505 Vandœuvre-lès-Nancy, France

**Keywords:** silver nanoparticles, physicochemical properties, antibacterial activity, methods of evaluation

## Abstract

Due to the constant increase in the number of infectious diseases and the concomitant lack of treatment available, metallic nanoparticles (e.g., silver nanoparticles) have been of particular interest in the last decades. Indeed, several studies suggest that silver nanoparticles have valuable antimicrobial activities, especially against bacteria, which may lead us to think that these nanoparticles may one day be an attractive therapeutic option for the treatment of bacterial infections. Unfortunately, when we look a little closer to these studies, we can see a very great heterogeneity (e.g., in the study design, in the synthetic process of nanoparticles, in the methods that explore the antibacterial properties of nanoparticles and in the bacteria chosen) making cross-interpretation between these studies impossible, and significantly limiting the interest of silver nanoparticles as promising antibacterial agents. We have selected forty-nine international publications published since 2015, and propose to discuss, not the results obtained, but precisely the different methodologies developed in these publications. Through this discussion, we highlighted the aspects to improve, or at least to homogenize, in order to definitively establish the interest of silver nanoparticles as valuable antibacterial agents.

## 1. Introduction

The 3rd of April 2018, the Centers for Disease Control and Prevention published on its official website an article about “nightmare bacteria”. These bacteria are resistant to all antibiotic tested, and more than 200 cases were reported in USA in 2017 [1]. It is urgent to react. New therapeutic strategies have to be proposed. These last fifteen years, the studies about using “nanoparticles” as antibacterial agents did not cease [2]. What about metallic nanoparticles as new antibacterial drugs?

The antimicrobial properties of several metals, such as platinum, copper, gold and silver are very widely documented in the scientific literature and some (i.e., gold and silver) are even known since Antiquity. Even if a substantial quantity of scientific literature covers the use of colloidal suspensions of metallic nanoparticles—mainly silver—only a few deals with the study of the influence of nanosized silver-core. Indeed, in large part of studies, the extract-plant or biosynthesis approach is used for the synthesis of the silver nanoparticles. As such chemicals used may have antibacterial activity, it is not possible to conclude about silver nanoparticles (Ag NPs) one. Moreover, as silver exhibit antibacterial properties, a large part of studies are devoted to the synthesis and characterization of silver-based-nanocomposites.

Hence, one question remains: are metallic nanoparticles and, more specifically, silver nanoparticles a promising antibacterial agent? To answer, it is necessary to consider studies dealing with the synthesis and antibacterial activity of aqueous colloidal solutions of silver nanoparticles obtained by a synthetic approach, which does not imply antibacterial stabilizers, biosynthesis, extract plant synthesis, composites and assimilated ones. Moreover, the colloidal solutions of silver nanoparticles have to be well-characterized, and the antibacterial activity has to be evaluated accurately (i.e., using standard procedure and identified bacteria strains). However, it is not always the case in reported studies. Therefore, we selected international publications from 2015 to 2018, referring to “antibacterial”, “silver” and “nanoparticles” keywords (using SciFinder^n^, we obtained 4779 articles published in English) and removed all ones dealing with biosynthesis/biogenic (green, plants, extract, leaf, bacteria, fungi, etc.), composites (coated/coating, hybrid, alloys, core-shell, bi-metallic, decorated, combined to other metals such as titanium, copper or iron, grafted, filer, doped, modified, porous, hydrogel, etc.), carbonaceous materials (carbon nanotubes, graphene, etc.), NPs with a mean size higher than 100 nm (such as fibers, etc.), dental adhesive or supported/embedded nanoparticles (cellulose, silk, etc.) and all formulations using silver nanoparticles combined with antibacterial agent or active molecules (chitosan, pectin, tannic acid, curcumin, etc.). Finally, we obtained forty-nine international publications published since 2015 (Table 1 and Appendix A). The aim of this review is to discuss the limits we are currently facing regarding the study of aqueous colloidal solutions of silver nanoparticles, in particular, the different methods used, from their synthesis to the evaluation of their antibacterial properties.

## 2. Ag NPs and Antibacterial Activity

### 2.1. Nanoparticles Definition

Following ISO/TS 80004-1:2015 recommendations, nano-objects are defined as a material, which has one, two or three dimensions less than 100 nm. As the antibacterial activity of nanoscale is under consideration, we have to define nanoparticles as a nano-objects having three dimensions ranging from 1 to 100 nm. Thus, metallic nanoparticles can be defined as a metal atoms aggregate—i.e., chemically bonded—exhibiting a certain size dispersity.

The LaMer’s theory can explain the formation of metallic nanoparticles in solution [52]. Briefly, in solution, metal ions condense to lower the system free energy. When the supersaturation in the precursor is reached, a critical germ forms. Then, the nanoparticle will be obtained by the growth of this critical germ by either adding an atom to the critical germ or by coalescence with another critical germ/nanoparticle. As the system wants to minimize its free energy, it will continue to grow until the formation of the corresponding bulk material. So, it is necessary to add a stabilizer (i.e., ions/molecule in interaction with nanoparticle surface) to avoid the aggregation process.

Thus, the preparation of NPs colloidal solution implies a solvent, a metal source, a reductant and a stabilizer. Several recent review papers describe the different strategies to design NPs, more specifically silver ones for biomedical applications [53,54,55,56,57,58]. In some cases, a unique compound can be used as solvent, reductant and stabilizers or as reductant and stabilizer—e.g., polyol process or citrate-based approach, respectively [59].

Therefore, stabilizer-NPs are characterized by their metallic-core-size, shape and structural properties, the nature of the stabilizer/capping agent. However, as we are interested in colloidal solutions for an antibacterial purpose, other parameters such as pH, ionic strength and all the chemicals present in the solution have to be considered.

### 2.2. Antibacterial Mechanism of Silver Colloidal Solutions

There are four principal possible mechanisms of silver colloidal solutions (Figure 1): (i) the formation of free radicals (e.g., reactive oxygen species) by redox reaction, (ii) Ag NPs adhesion to bacteria cell membrane inducing its destabilization, (iii) the intercalation of Ag NPs between DNA bases inhibiting DNA replication and transcription and (iv) Ag NPs-induced ribosome destabilization inhibiting protein synthesis. The antibacterial mechanism of silver colloidal solutions is however not elucidated: Ag NPs antibacterial activity results from Ag^+^ ions release from Ag NPs or the Ag NPs themselves or both. For more details about these aspects, we recommend the following references [60,61,62,63,64,65].

### 2.3. Colloidal Solution Parameters Influencing Antibacterial Activity

The two parameters that we can find in almost all the studies dealing with Ag NPs and the antibacterial activity are the size and the shape of the nanoparticles (Table 1). However, there are other parameters of prime importance for antibacterial activity such as stability and accessibility of NPs’ surface, and silver concentration.

#### 2.3.1. NPs-Metallic-Core Size

The NPs-metallic-core size is usually considered as the main parameter because as the size decreases the surface-to-volume ratio increases resulting in physicochemical and biological properties differing from the corresponding bulk material. Indeed, a small size will facilitate the interaction with the bacteria surface and will potentially have higher antibacterial activity. For instance, Ajitha et al. prepared, from silver nitrate, polyvinyl alcohol (PVA), and NaBH_4_ in water, spherical PVA-Ag NPs [26]. They obtained four PVA-Ag NPs sizes by varying the pH of the solution with NaOH (from SEM: 31, 24, 19 and 14 nm at pH = 6, 8, 10 and 12, respectively). Then they evaluated their antibacterial activities using the Kirby-Bauer method and *Escherichia coli* and *Pseudomonas* spp. as test bacteria. The data obtained indicate that the inhibition disk size increases when the NPs size decreases. However, as NPs samples possess, by definition, a size distribution it is quite difficult to give a definitive conclusion, and many questions remain: Which proportion of each NPs sizes are present? Which one is more efficient? As the presence of Ag_2_O leads to Ag^+^ ions release in solution [66], what is the chemical composition and oxidation state of Ag NPs (i.e., Ag° or Ag° @ Ag_2_O)?

#### 2.3.2. Shape and Structural Properties

During their formation, NPs minimize their total free energy by adopting facetted structures. These structures exhibit faces of different Miller indices, mainly (111) and (100) faces. It worth to note that (111) faces have lower free energy than (100) ones (i.e., γ (111) < γ (100)). Hence, the so-called “spherical” NPs possess mainly (111) faces whereas others kind of NPs (e.g., nanoplates, nanorods) present large (100) faces. The NPs shape and the kind of faces will also influence bacteria/NPs interaction and then their antibacterial activity. For instance, Hong et al. added NaCl to the reaction medium to modify the nanoparticles shape [34]. Starting from AgNO_3_ and polyvinylpyrrolidone (PVP k30) in ethylene glycol, they obtained “spherical” nanoparticles of 60 nm (±15 nm) exhibiting mainly {111} facets. With a small amount of NaCl, truncated-nanocubes are formed. With these truncated-nanocubes, there are by-products such as bipyramids, nanospheres and nanorods. The average size of these truncated-nanocubes is 55 nm (±10 nm), and the sample exhibits mainly {100} facets with {111} ones for truncated faces. After treatment of *E. coli* with Ag NPs, they found that {100} facets structures have higher antibacterial activities than ones with {111} facets. It can be attributed to higher reactivity of the {100} facets, which have higher energy than {111} facets. Therefore, nanocubes attach more rapidly to the bacteria cell membrane, and they also have a higher contact surface with bacteria membrane in comparison with spherical nanoparticles. This last point explains why with nanowires (length >2–4 μm) they obtained had the lowest efficiency. Moreover, various nanoparticles shapes can also be achieved with the seed-mediated growth approach [67]. Hu et al. used seed-mediated growth to prepare silver nanorods starting from 2 nm seeds and cetyltrimethylammonium bromide (CTAB) and ascorbic acid [68]. By varying the number of seeds in the growth solution, they obtained as main products nanospheres or nanorods (spheres mean size: 42 nm; rods mean length: 70 and 85 nm). The antibacterial activities have been evaluated by treating *S. aureus* and *E. coli* with the different samples of NPs, but no significant difference can be observed. However, as CTAB exhibits antibacterial activity [69,70], it is not evident to decide between the influence of CTAB or NPs shape on their antibacterial activities. Thus, even if NPs shape and structure should have a strong influence on antibacterial activity, the lack of structure characterization limits our comprehension of its influence. Moreover, if different capping agents and/or different average sizes are used, it becomes also difficult to conclude on the influence of these parameters on antibacterial activity.

#### 2.3.3. Surface Stability

The surface stability of NPs is related to colloidal solution one. In a non-stable colloidal solution, the NPs tend to aggregate and so reducing the accessible surface area. Therefore, the potential zeta and capping agent are also two critical parameters.

A colloidal solution is usually considered as stable when the zeta potential is lower or higher of −30 or +30 mV. As bacteria membrane cells are negatively charged, a colloidal solution having positive zeta potential should have enhanced antibacterial activity. In most of the cases, the reported zeta potential values are obtained from a suspension of nanoparticles in water. Are these values always valid in culture media? If yes, we can correlate its sign/value to antibacterial activity. However, metallic nanoparticles exhibit surface plasmon resonance (SPR), and this phenomenon can be characterized by UV-vis spectroscopy. As the SPR depends on the surface electronic environment, the modification of the UV-vis spectrum of nanoparticles in water and culture medium observed is related to surface interaction. Hence, the zeta potential, which corresponds to the electric potential surrounding nanoparticle surface; should also be modified when the colloidal solution is added to culture medium. Moreover, most of colloidal solution used for antibacterial activity evaluation has a negative zeta potential (see Appendix A). To elucidate the influence of zeta potential, it is necessary to perform a series of measurements with different pH and ionic strength values close to culture media ones.

#### 2.3.4. Surface Accessibility

Capping agent at the surface of the metallic core is either preventing or inducing NPs aggregation [71]. Moreover, depending on the pH of the medium and surface dissociation constants of the capping agents, the NPs’ assembly can occur [72]. Furthermore, the presence of divalent cations can induce assembly of NPs by bridging stable negatively charged colloidal solutions [73]. However, capping agent influences also the interaction between medium/bacteria and NPs surface. As NPs-bacteria surfaces interaction seem to play a crucial role in antibacterial activity (e.g., penetration of NPs in the bacterial cell), it is essential to verify NPs surface accessibility when capping agent is grafted to its surface. This aspect is, however, most of the time not considered in the studies dealing with metallic NPs and antibacterial activities. A catalytic reaction can be used to prove the surface accessibility. For instance, the hydrogenation of 4-nitrophenol by NaBH_4_ in water is possible only in the presence of metallic nanoparticles (e.g., silver or gold nanoparticles) [74]. The systematic use of this characterization of Ag NPs surface could give information about surface reactivity and one point of comparison. However, the reactions have to be conducted using the same experimental conditions (i.e., same concentrations, temperature, the proportion of nanoparticles, etc.) to facilitate the comparison between studies.

#### 2.3.5. Other Chemicals and Concentration of Silver

It is also important to consider all chemicals used for the preparation of the NPs colloidal solutions. Indeed, in some cases, molecules having recognized antibacterial activity are used during the NPs synthesis, as reductant or solvent for example [32,46]. In such a case, it is no longer possible to consider only stabilizer-Ag NPs, but antibacterial molecule activity has also to be.

Moreover, if the antibacterial activity of all chemicals used is not known, they also must be evaluated. Therefore, it is crucial to have reference tests with the metallic precursor, stabilizer and all other chemicals used for NPs synthesis. Moreover, the activity of metallic NPs has to be compared to at least one “conventional antibacterial agent” in the same conditions.

The silver concentration is also an important parameter, which has to be considered and can be determined by different techniques, such as inductively coupled plasma mass spectrometry (ICP-MS). Hence, the comparison of the minimal inhibitory concentration (MIC) between different studies could be facilitated by expressing MIC in molar concentration (mol/L). Moreover, as silver-ions-antibacterial activity is known since centuries, it is crucial to determine silver ions concentration in colloidal solution and culture media after incubation time.

## 3. How to Evaluate the Antibacterial Properties of Ag NPs?

Most of the bacteria are divided into two groups: Gram-positive bacteria and Gram-negative bacteria. This distinction is primarily based on the structure of the bacterial cell wall (Figure 2).

This difference in cell wall structure is significant because it has a direct impact on the activity of certain antibiotics that are then called narrow spectrum, i.e., only active on Gram-negative bacteria or Gram-positive bacteria.

For several years we have been confronted with a particularly worrying phenomenon: resistance to anti-infectives. This resistance is found in all microorganisms, to different degrees, but is particularly worrying in bacteria. Indeed, bacteria have acquired the formidable ability to develop mechanisms of resistance to anti-infectives, making most of the antibiotics used in therapy obsolete. This situation is particularly worrying given the limited number of new antibacterial drugs available, and continued emergence and rapid spread of newly resistant bacterial strains. Therefore, there is an urgent medical need for new antibacterials [75].

In the search for new antibacterial drugs many studies have sought to study the antibacterial properties of metal nanoparticles in general, and silver nanoparticles in particular.

### 3.1. Which Bacteria Test?

If we want to determine the antibacterial spectrum of Ag NPs, it is, therefore, necessary to test their effectiveness on several different bacteria and if possible both on Gram-positive and Gram-negative bacteria. In most studies, there are two major bacterial species: *Escherichia coli* and *Staphylococcus aureus* (Table 1 and Appendix A). Indeed, these two species are those most frequently encountered in human infectious diseases. They are also particularly resistant to antibiotics and therefore problematic [76,77]. Now for the same species, there are several strains, and these strains all have different properties. It is difficult to compare studies that have not been conducted on the same strains of a species. This may lead some authors to different or even opposite conclusions.

The choice of the bacterium to be tested is also fundamental depending on the therapeutic applications that one wishes to develop for the use of Ag NPs. We can notably quote the study of Niska et al. who wanted to study the interest of the Ag NPs in the oral field and who thus chose to work on bacteria of the oral commensal flora [48].

However, some bacterial species have no interest in human infectious diseases; we think in particular to *Bacillus subtilis*. Most of the time, these authors use this bacterium as a model, which may be conceivable, but then, it would be more scientifically right to be very careful about the interpretation of the results, e.g., by recalling that this bacterium does not have interest in human disease, which de facto would limit the interest of the study itself.

Many authors study the antibacterial properties of Ag NPs either on strains from a collection (e.g., ATCC) or clinical isolates (Table 1 and Appendix A). The interest of strains from a collection is that they are perfectly characterized (phenotypically, antibiotic resistance profile), which is rarely the case for clinical isolates, but these ones are directly isolated from patients and therefore correspond to a unique pathological situation (normally) clearly defined. It should be noted, however, that some studies do not specify the origin of the bacteria that have been tested [19,20,26,27,31,42,78], and this is hardly acceptable from a scientific point of view.

Now if we want at a time to demonstrate the full interest of Ag NPs, we will evaluate their antibacterial properties on clinical isolates, on bacteria resistant to antibiotics. Unfortunately, if we look at Table 1 and Appendix A, finally very few studies have been done on antibiotic-resistant bacteria; and in these studies, we rarely have the antibiograms of the bacterial strains studied. This is the next step that we must take if we want to prove the interest of a potential use of Ag NPs in human (or animal) therapy.

### 3.2. Which Technique to Use?

There is currently no standard, no reference for studying the antibacterial activities of nanoparticles. Moreover, as we can see (Table 1 and Appendix A) several techniques, very different from each other, are used by the authors to evaluate the antibacterial properties of Ag NPs. For a review of the different techniques that exist to evaluate the antibacterial properties of molecules, we recommend the review by Balouiri and colleagues [79].

However, this poses a first problem: from the moment when the studies do not use all the same technique, it will be challenging, if not impossible, to compare the results. In the laboratory, we have chosen to adapt the method used for antibiotics: microdilution in liquid medium [80]. Indeed, after evaluating several protocols, this technique allows us: not to have any problem of diffusion of Ag NPs in the culture medium (which is not the case with the agar method), it is possible to agitate the microplate to promote contacts between bacteria and Ag NPs (which is not possible with methods using Petri dishes), this technique is the only one that can give us a minimum inhibitory concentration correct (which must be the first step when conducting a study on the antibacterial properties of molecules).

Once the problem of the method is settled, another question arises: which culture medium to use? Again, if we look at the different studies (Table 1 and Appendix A), there is no homogeneity in the choice of culture media used. On the contrary, here again, a very large number of culture media, very different from each other are used: Luria Bertani, Mueller Hinton and Nutrient Agar. This point can seem anecdotal for some authors, but it is on the contrary fundamental. Bacteria are living organisms, and each culture medium has a different substrate composition; in such a way that, depending on the culture medium used for the experiments, we will not have the same growth kinetics of the bacteria, nor the same phenotypic characters for these bacteria. In the end, if the growing conditions do not allow us to control the growth of bacteria how to be certain that we will only evaluate the antibacterial properties of NPs? Again, it is not possible.

In the laboratory, we chose to use the cation-adjusted Mueller Hinton medium. The cation-adjusted Mueller Hinton medium is the only one recommended for performing antibiograms in clinical bacteriology [81]. Historically it has been developed for positively charged antibiotics (e.g., colistin) which is still somewhat interesting in the case of Ag NPs. It is also important to specify that this medium should be complemented in the case where the study focuses on demanding or “fastidious” bacteria (i.e., that require an additional substrate for their growth, such as adding blood for streptococci, for example) [81].

In the end, as we have just seen, it is difficult to conclude on the results of the studies we have selected, although there is evidence on the antibacterial activities of Ag NPs. However, what is even more difficult is to compare studies with each other. Nevertheless, we can highlight the efforts that have been made by some authors, we can for example highlight the work of Cavassin and collaborators or Niska and collaborators [44,48] who worked on several bacterial species and for some species several strains were tested, using accurate techniques; proof that we progress and we have to continue this way.

## 4. Conclusions and Future Perspectives

In the fight against infectious diseases, the report is alarming: we are currently very badly engaged, microorganisms are gaining, day by day, more battles. There are several reasons for this situation, and we will remember two: antimicrobial resistance and the urgency to find new anti-infectives [75]. One of the research topics in our research laboratory is dedicated to the research of new anti-infectives. Moreover, we believe, like other numerous authors, that metallic nanoparticles, such as silver nanoparticles, might represent, in the future, a therapeutic option for the management and the treatment of infectious diseases. In any case, silver nanoparticles (and more generally metallic nanoparticles) deserve our interest in them; that is the challenge. This review aimed to take stock, through the latest publications on the subject, about the different methodologies that are currently used, considering the synthesis of nanoparticles, their characterization, and even the evaluation of their antibacterial properties. The idea was to be able to compare the studies with each other, to have the best judgment regarding the promising use of Ag NPs. In the end, it appears that it is impossible to compare the studies between them: there are too many discrepancies between studies (Table 1 and Appendix A). Indeed, we think that it is urgent to homogenize the studies, to better characterize the nanoparticles, to choose a method (or to limit their number) for the evaluation of the antibacterial properties, to choose bacteria of interest, in agreement with the epidemiological data or the therapeutic uses envisaged for the silver nanoparticles. It is at this price that the data will have more impact on the whole of the scientific community (Table 2).

Nevertheless, we must also point out a critical feature of Ag NPs: the toxicity. Indeed, having an excellent antibacterial agent, with very interesting MICs, is an essential first step, when you are looking for the antibacterials of tomorrow. However, at the same time, you must never forget the second and equally important step: the safety of your “promising” antibacterial agent (i.e., Ag NPs). De facto, many studies have reported a possible toxicity of Ag NPs, both in vivo and in vitro (see [82] for a state-of-the-art review). Finally, studying the mechanisms of interaction between Ag NPs and biological cells (i.e., eukaryotic cells) in order to better appreciate the potential risks related to a possible future use of Ag NPs as antibacterial agents, also seems to become a significant issue.

## Figures and Tables

**Figure 1 nanomaterials-09-01775-f001:**
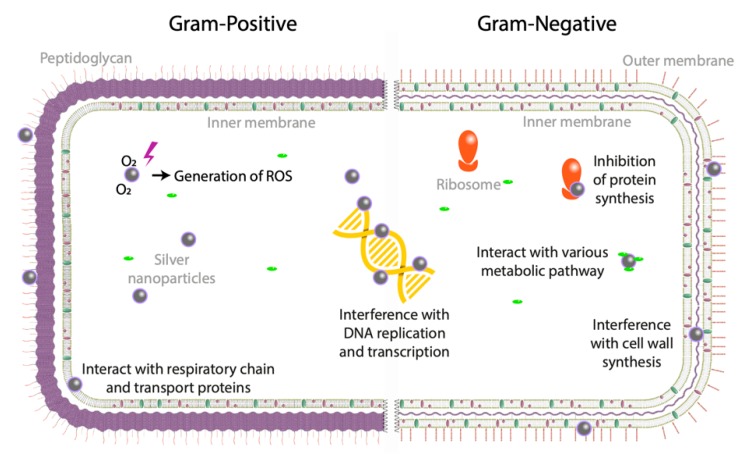
Probable antibacterial mechanisms of action of silver nanoparticles.

**Figure 2 nanomaterials-09-01775-f002:**
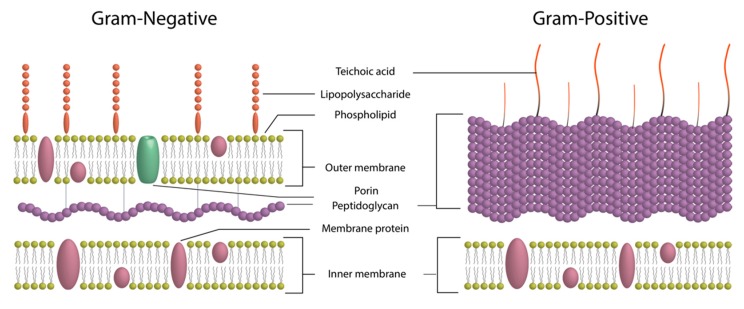
Bacterial cell wall structures.

**Table 1 nanomaterials-09-01775-t001:** Methods, synthesis and characterizations of silver nanoparticles, and antibacterial activity evaluation of selected publications from the 2015–2018 period.

Stabilizer	NPs Size (nm)	NPs Shape	Stock Suspension Concentration or Mass	Zeta Potential (mV) *	Comment	Bacteria	Bacteria Origin	Protocol	Culture Media	Reference
“Naked”	19.5 ± 7.7	Nano-sphere	N/A	−18.0 ± 0.6 (in culture media)	NaBH_4_ + AgNO_3_ in presence of ultrasonication	*E. faecalis* *S. aureus* *S. epidermidis* *B. subtilis* *E. coli* *S. typhimurium* *S. enterica*	KCCM 13807 KCTC 1916 KCTC 1971 KCTC 1021 KCTC 1682 KCCM 40253 KACC 10763	Kirby-Bauer method Microdilution method MIC90	Mueller Hinton Agar Luria Bertani	[3]
	10	Nano-flake	50 ppm	N/A	Axonnite Silver suspended in demineralized water	*A. baumannii* (*n* = 17) *A. baumannii* *A. nosocomialis* (*n* = 10)	Clinical isolates ATCC 1906 Clinical isolates	Microdilution method	N/A	[4]
	10 20 40	Nano-sphere	N/A	N/A	Vapor nucleation in N_2_ gas	*E. coli*	N/A	Kirby-Bauer method	Mueller Hinton 2 Agar	[5]
	55.6 ± 2.9 (DLS)	Nanosphere	8.53 mg	−51.5 ± 2.5	Electrochemically synthesized	*P. aeruginosa* (*n* = 3) *S. maltophilia* (*n* = 3) *B. cepacia* (*n* = 3) *S. aureus* (*n* = 3)	Clinical isolates	Kirby-Bauer method Microdilution method TKA	Mueller Hinton Cation-Adjusted	[6]
	2–5 (70–75% TEM)	Nano-sphere	50 mg/kg	+9.2	Nano-Tech (Warsaw, Poland)	*L. monocytogenes*	PCM 2191	Microdilution method	Tryptone Soy yeast extract broth	[7]
	23.6 (TEM) 57.8 (DLS)	Nano-sphere	N/A	−28.3 (N/A)	Nanoleader (Korea)	*E. coli* (K-12)	KCTC 1116	Growth Curves	Luria Bertani	[8]
	10	Nano-sphere	1 mg/mL	N/A	Sisco Research lab.	*S. aureus* *S. aureus* (*n* = 30)	ATCC 25923 Clinical isolates	Agar Well diffusion method Microdilution method	Mueller Hinton Agar Luria Bertani	[9]
“Naked”	40 (TEM)	Nano-sphere	N/A	N/A	AgNO_3_ + NaBH_4_	*S. aureus* *B. cereus* *P. aeruginosa* *K. pneumoniae* *E. coli*	N/A	Agar Well diffusion method	Mueller Hinton Agar	[10]
Unknown “Naked”	35 27.2	Nano-sphere	20 µg/mL N/A	N/A	Sigma Aldrich (Dorset UK) Laser generated	*E. coli* (K-12)	JM 109	Agar Well diffusion method	Mueller Hinton	[11]
Citrate	23 ± 2 (TEM)	Nano-sphere	2 mM	N/A	Citrate BioPure™ Silver, Nanocomposix (San Diego, CA, USA)	*L. monocytogenes* (*n* = 20) *L. monocytogenes*	Clinical isolates	Colony Forming Units	Mueller Hinton	[12]
	6.0–28.2 (XRD)		N/A	−28.2 to −32.0	Plasma discharge	*S. aureus* *E. coli*	ATCC 25923 ATCC 35218	Kirby-Bauer method	Nutrient Agar	[13]
	2.3 ± 0.5 12.5 ± 2.2 32.4 ± 6.5 (TEM)	Nano-sphere	N/A^1^	N/A^1^	NaBH_4_ + AgNO_3_ + sodium citrate	*S. aureus* *E. coli*	ATCC 35696 ATCC 23282	Microdilution method Kirby-Bauer method Growth Curves	Broth medium	[14]
	40–50 20 × 20–90	Nano-sphere Nano-rod	N/A	−28.8 −23.5	Citrate thermal reduction method	*S. aureus* *B. subtilis* *P. aeruginosa* *K. pneumoniae* *E. coli*	ATCC 25923 AST5-2 AL2-14B AWD5 ATCC 25922	Microdilution method Kirby-Bauer method TKA	Nutrient Agar	[15]
	20.1 ± 4.4 (TEM) 49.3 ± 5.7 (DLS)	Nano-sphere	N/A	−19.2 ± 0.7	Citrate thermal reduction method	*E. coli* *S. aureus* *L. bulgaricus* *L. casei*	ATCC 25922 ATCC 25923 CGMCC 1.6970 CGMCC 1.2435	TKA	Luria Bertani Tryptone Soy MRS	[16]
	10–40 (TEM)	Sharp-tipped triangular, truncated triangular, nanoprisms, decahedra, tetrahedra	N/A	N/A	Photochemical synthesis: 4.76 ± 3.88 nm silver seed nanoparticles (AgNO_3_ + citrate + NaBH_4_) + 40 W blue LEDs (Hongke Lighting kem = 455–475 nm) {111} facets	*E. coli*	ATCC 25922	Growth Curves	Luria Bertani	[17]
Citrate	20 ± 9 25 ± 3 11 ± 6 (TEM)	Nano-sphere	N/A	−26.37 −37.95 −28.23 (H_2_O + salts)	AgNO_3_ + citrate + NaBH_4_	*S. aureus*	ATCC 25923	Agar Well diffusion method	Nutrient Agar	[18]
	42–58 (TEM)	Nano-sphere	N/A	N/A	AgNO_3_ + citrate + NaBH_4_; polydisperse; XRD11: intense reflection at (111)	*S. aureus* *S. pyogenes* *S. typhi* *P. aeruginosa*	N/A	Kirby-Bauer method	Tryptone Soy	[19]
	15 18 30 30 (DLS)	Nano-sphere	N/A	−38.8 −30.7 −38.5 −42.2 (N/A)	AgNO_3_ + Citrate + NaBH_4_	*E. coli* *B. subtilis*	N/A	Kirby-Bauer method	Nutrient Agar	[20]
GSH13	10–50	Nano-sphere	0.197 mg/mL	N/A	AgBF_4_ + NaBH_4_ + glutathione	*C. jejuni* (*n* = 22) *C. coli* (*n* = 18) *C. jejuni*	Animal or Human clinical isolates NCTC 11168	Microdilution method	Brucella Mueller Hinton	[21]
D-xylose L-arabinose D-ribose D-glucose D-galactose D-mannose D-lactose D-xylose	33 30 39 25 28 25 15 18	Nano-sphere	N/A	N/A		*E. coli* *Klebsiella* spp.	N/A	N/A	N/A	[22]
L-fucose	10.15 ± 3.37 (TEM)	Nano-sphere	N/A	−65.4 −17.7	AgNO_3_ + NaBH_4_ + sodium citrate + mercaptopropionic acid, then L-fucose	*P. aeruginosa* (PAO1)	N/A	Microdilution method	Luria Bertani without Chloride ions	[23]
PEG	15.8 ± 2.2 (TEM)	Nano-sphere	N/A	−17.2 ± 2.1	AgNO_3_ + EG/PEG	*S. aureus* *P. aeruginosa* *S. enterica* *E. coli*	ATCC 6538 ATCC 15442 ATCC 10708 ATCC 11229	Microdilution method	Mueller Hinton	[24]
PC	3.3 ± 0.9 4.9 ± 2.9 (TEM)	Nano-sphere	N/A	N/A	Zwitterionic Protection: AgNO_3_ + NaBH_4_ + PC-SH	*E. coli* *S. aureus*	OW6 Mu50	Growth curves	Todd Hewitt broth	[25]
PVA	31 (SEM; TEM: 26) 24 (SEM) 19 (SEM) 14 (SEM; TEM: 10)	Nano-sphere	N/A	N/A	PVA + AgNO_3_ + NaBH_4_ (pH = 6; 8; 10; 12, respectively)	*E. coli* *Pseudomonas* sp.	N/A	Kirby-Bauer method	Nutrient Agar	[26]
PVP/citrate	50–60 70–80	Semi-triangular and truncated triangular silver nanoparticles + few nano-sphere Triangular silver nanoparticles with sharp corner	N/A	N/A	Citrate + AgNO_3_ + NaBH_4_ + PVP + visible-light halogen lamp (50 and 100 W, respectively); bigger nanoparticles (>100 nm) obtained with visible-light halogen lamps with higher intensities	*E. coli*	N/A	Colony Forming Units	Nutrient Agar	[27]
PVP^20^	14.0 ± 0.3 (TEM)	Nano-sphere	1 mg/mL	−27.3	Nanocomposix, OECD standard BioPure, PVP^20^ 40kDa	*E. coli* (K-12) *B. subtilis*	MG1655 ATCC 6051	Growth curves	Tryptone Soy	[28]
	5 20	Nano-sphere	1 mg/mL 2 mg/mL	N/A	Shanghai Institute of Fine Chemical Materials (China)	*E. coli* *P. aeruginosa* *S. aureus* *S. epidermidis*	ATCC 8739 ATCC 9027 ATCC 6538 ATCC 12228	Poisoned Food Technique	Mueller Hinton	[29]
	8 29 (TEM)	Nano-sphere	N/A	−22.36 −37.82	PVP + ethylene glycol + AgNO_3_; redispersed in water	*A. hydrophila* *P. putida* *E. coli* *B. subtilis* *S. aureus*	4AK4 KT2442 Trans 1-T1 ATCC 28357 N/A	Kirby-Bauer method Microdilution method	N/A Mueller Hinton	[30]
	15.6 (TEM)	Nano-sphere	N/A	N/A	PVP + AgNO_3_ + NaBH_4_ in water; polydisperse	*Citrobacter* sp. *Enterococcus* sp.	N/A	Colony Forming Units	N/A	[31]
	3–34 (TEM)	Nano-sphere	N/A	N/A	AgNO_3_ + Ethanol + PVP (55000 molecular mass) in water	*S. aureus* PTCC No. 1112 *E. coli* PTCC No. 1330	ATCC 6537 ATCC 8739	N/A	Mueller Hinton	[32]
	10–15 (TEM)	Nano-sphere	N/A	N/A	AgNO_3_ + PVP (k30; Mw. 40000) + hydrazine; influence of AgNO_3_, PVP and hydrazine concentrations	*E. coli* *S. aureus*	ATCC 25922 ATCC 25923	Kirby-Bauer method	Nutrient Agar	[33]
	60 ± 15 55 ± 10 60 × 2000–4000	Nano-sphere Nano-cube, right bipyramids Nanowire	N/A	N/A	AgNO_3_ + PVP (k30) + ethylene glycol + NaCl (0. 1, and 5 mg, respectively)	*E. coli*	ATCC 25922	Growth curves Microdilution method	Luria Bertani	[34]
	20.6 ± 3.1	Nano-sphere	1 mg/mL	−35	Nanocomposix (San Diego, CA, USA)	*C. jejuni* (*n* = 4) *C. jejuni* *Salmonella* spp. (*n* = 5)	Chicken isolates NCTC 11168 Chicken isolates	Microdilution method	Mueller Hinton Luria Bertani	[35]
PVP Glycerol	31.2 (TEM) 46.5 (DLS)	Nano-sphere	N/A	+18.7 (N/A)	AgNO_3_ + PVP or glycerol + sodium citrate (NH_4_OH, pH = 8)	*C. sakazakii*	ATCC 29544 ATCC BAA894 ATCC 29004 ATCC 12868	Microdilution method Oxford cup method	Luria Bertani	[36]
Oleylamine	10 (TEM)	Nano-sphere	N/A	−7.11	ColdStones Tech. (Suzhou, China)	*B. subtilis*	ATCC 6633	Growth curves	Luria Bertani	[37]
Casein	12.5 ± 4 (TEM) 50.0 ± 0.7 (DLS)	Nano-sphere	N/A	−26.6 ± 1.7 {in 3-(N-morpholini) propanesulfonic acid-(hydroxy-methanyl) aminomethane}	Lab. Argenol S. L. (Zaragoza, Spain)	*E. coli* *P. aeruginosa*	MC 1061 DS 10-129	Bioluminescence inhibition assay	Luria Bertani	[38]
Sericin	3.78 ± 1.14 (TEM)	Nano-sphere	N/A	N/A	Silk sericin protein + AgNO_3_ + NaBH_4_	*S. aureus* *E. coli*	ATCC 25923 ATCC 25922	Cell counting (FCM)	Nutrient medium	[39]
Thioacetic acid Propionic acid	20–25 30–35	Nano-sphere	N/A	N/A	Thioacetic or propionic acid + silver acetate + sodium carbonate (US20120100372A1)	*S aureus* *S. epidermidis* *A. baumannii* *P. aeruginosa*	ATCC 25923 ATCC 35984 ATCC 19606 ATCC 27853	Microdilution method	Mueller Hinton	[40]
Lipoid acid	2.0 ± 0.5 (TEM)	Nano-sphere	N/A	N/A	dihydrolipoic acid + NaOH + AgNO_3_ + NaBH_4_	*S. aureus* *E. coli* *E. coli*	N/A DH5α DSM4230	Growth curves	Luria Bertani	[41]
PEG EDTA PVP PVA	44 39 35 31 (SEM)	Nano-sphere	N/A^1^	−17.5 −23.0 −41.0 −47.0	PEG, EDTA, PVP or PVA + AgNO_3_ + NaOH + NaBH_4_ in water	*E. coli* *Pseudomonas* spp.	N/A	Kirby-Bauer method	Nutrient Agar	[42]
“Naked” Unknown	7.5 10.1 (TEM)	Nano-sphere	9.7 × 10^−8^ mol/L 4 × 10^−8^ mol/L	−38.0 0.0	AgNO_3_ + NaBH_4_ Rice starch + AgNO_3_	*S. aureus* *S. mutans* *S. pyogenes* *E. coli* *P. vulgaris*	ATCC 29737 ATCC 35668 ATCC 8668 ATCC 15224 ATCC 7829	Agar Well diffusion method Microdilution method	Brain Heart Infusion Mueller Hinton	[43]
PVA Citrate Citrate	10 nm (SEM) 40 nm (SEM) 60 nm	Nano-sphere	N/A N/A 20 μg/L	−17.0 −48.4 N/A	Chitosan-Ag NPs also prepared and tested (Sigma Aldrich)	*A. baumanni* (*n* = 17) *P. aeruginosa* (*n* = 13) *Enterobacteriaceae* (*n* = 21) *S. maltophilia* (*n* = 2) *S. aureus* (*n* = 13) *S. aureus* *S. epidermidis* *Enterococcus* sp. (*n* = 14)	Clinical isolates Clinical isolates Clinical isolates Clinical isolates Clinical isolates ATCC 29213 INCQS 198 Clinical isolates	Agar Well diffusion method Microdilution method Time kill assay	Mueller Hinton Mueller Hinton Cation Adjusted Tryptone Soy Mueller Hinton Cation Adjusted	[44]
Cysteine PVP	7.6 ± 1.5 7.7 ± 1.6 (TEM)	Nano-sphere	N/A	N/A	AgNO_3_ + NaBH_4_ + L-cysteine AgNO_3_ + KOH + PVP (Mw 8000) + NaBH4	*S. aureus* *E. coli* *P. aeruginosa*	ATCC 29213 ATCC 23716 ATCC 25619	Microdilution method	Mueller Hinton Cation Adjusted	[45]
Citrate MPA MHA MPS	10.2 ± 2.3 10.2 ± 2.5 10.2 ± 2.2 9.9 ± 2.0	Nano-sphere	N/A	−47.4 −34.5 −29.1 −32.6	Citrate + tannic acid + AgNO_3_ Citrate-Ag NPs + mercaptopropionic acid (ligand exchange) Citrate-Ag NPs + mercaptohexanoic acid (ligand exchange) Citrate-Ag NPs + mercaptopropionic sulfonic acid (ligand exchange)	*E. coli*	MG 1655	Growth curves	Luria Bertani	[46]
Citrate PVP PEG	40 (TEM; 10–70)	Nano-sphere Triangular silver nanoparticles with rounded edges Hexagonal silver NPs	N/A	N/A	AgNO_3_ + citrate PVP (Mw = 25000) + AgNO_3_ + citrate + H_2_O_2_ + NaBH_4_ PEG (2000) + NaOH + AgNO_3_	*E. coli*	DH5α	Agar Well diffusion method Growth curves	Luria Bertani	[47]
Lipoid acid PEG “Naked”	9.5 ± 1.9 (TEM) 9.8 ± 2.0 11.2 ± 2.1	Nano-sphere	N/A	−28.6 (Serum free culture medium) −10 (Serum free culture medium) −33.9 (Serum free culture medium)	Nanocomposix, Europe Nanocomposix, Europe *Also Tannic acid-NPs* US research (Nanomaterilas (Houston, TX, USA)	*Actinomyces* (*n* = 1) *Bacteroides* (*n* = 4) *Bacteroides fragilis* *Bifidobacterium* (*n* = 1) *Bifidobacterium breve* *Finegoldia* (*n* = 2) *Fusobacterium* (*n* = 4) *Fusobacterium nucleatum* *Parabacteroides* (*n* = 1) *Parvimonas* (*n* = 2) *Peptostreptococcus* (*n* = 1) *Peptostreptococcus anaerobius* *Porphyromonas* (*n* = 3) *Porphyromonas levii* *Prevotella* (*n* = 5) *Prevotella loescheii* *Propionibacterium* (*n* = 2) *Tannerella* (*n* = 1) *S. aureus* *S. aureus* *S. aureus* *S. epidermidis* *S. mutans*	Clinical isolates Clinical isolates ATCC 25285 Clinical isolates ATCC 15700 Clinical isolates Clinical isolates ATCC 25585 Clinical isolates Clinical isolates Clinical isolates ATCC 25286 Clinical isolates ATCC 29147 Clinical isolates ATCC 15930 Clinical isolates Clinical isolates ATCC 25923 ATCC 6538 ATTC 6538P ATCC 14990 ATCC 29175	Plate dilution method Microdilution method	Brucella agar supplemented Mueller Hinton	[48]
Citrate HH SHSH SHST	15 ± 4 13 ± 2 13 ± 4 10 ± 6 (TEM)	Nano-sphere	290 mg/L 290 mg/L 330 mg/L 330 mg/L	−39.8 ± 0.74 (H_2_O + NaCl) −34.0 ± 1.97 (H_2_O + NaCl) −34.4 ± 2.03 (H_2_O + NaCl) −35.9 ± 1.09 (H_2_O + NaCl)	AgNO_3_ + Citrate + NaBH_4_ Hydroxylamine hypochlorite + NaOH + AgNO_3_ Sodium hypophosphite + sodium hexametaphosphate + AgNO_2_ Sodium hypophosphite + sodium hexametaphosphate + sodium tripolyphosphate + AgNO_2_	*E. coli* (K-12) *E. coli*	ATCC 10798 ER2566	Microdilution method	Mueller Hinton	[49]
Starch	8 ± 4 (TEM)	Nano-sphere	N/A	N/A	AgNO_3_ + NaBH_4_ + (C_6_H_10_O_5_)_n_	*S. aureus* *E. coli*	N/A	Kirby-Bauer method	Nutrient agar	[50]
AOT	20 50	“Nano-sphere”	N/A	N/A	Bis(2-ethylhexyl) sulfosuccinate + AgNO_3_ + ascorbic acid	*E. coli* *S. aureus*	N/A	Microdilution method	Luria Bertani	[51]

N/A: not available; KCCM: Korean Culture Center of Microorganisms; KCTC: Korean Collection of Type Cultures; KACC: Korean Agriculture Culture Collection; MIC: minimal inhibitory concentration; ATCC: American Type Culture Collection; DLS: dynamic light scattering; TKA: Time Kill Assay; TEM: transmission electron microscope; PCM: Polish Collection of Microorganisms; XRD: X-ray diffraction; CGMCC: China General Microbiological Culture Collection Center; MRS: deMan, Rogosa and Sharpe medium; GSH: glutathione; NCTC: National Collection of Type Culture; PEG: polyethylene glycol; CTAB: cetyl-trimethyl ammonium bromide; NTA: nanoparticle tracking analysis; PC: phosphorylcholine; PVA: polyvinyl alcohol; SEM: scanning electron microscope; PVP: polyvinylpyrrolidone; ATCC: American Type Culture Collection; EDTA: ethylenediaminetetraacetic acid; FCM: flow cytometry; MPA: mercaptopropionic acid; MHA: mercaptohexanoic acid; MPS: mercaptopropionic sulfonic acid; HH: Hydroxylamine hypochlorite; SHSH: Sodium hypophosphite and sodium hexametaphosphate; SHST: sodium hypophosphite, sodium hexametaphosphate and sodium tripolyphosphate; AOT: Bis(2-ethylhexyl) sulfosuccinate; * in aqueous medium if not specified.

**Table 2 nanomaterials-09-01775-t002:** Proposed recommendations.

Nanoparticle Samples		
Synthesis	Specification of all chemicals used	[83]
Characterizations	Metal core size (TEM) Hydrodynamic size (DLS) Zeta-potential Stock suspension concentration in metal or metal mass (e.g., ICP-AES)	
Microbiology		
Antibacterial activity	Specification of the procedure used for antibacterial activity determination: Broth Dilution Procedures (Macrodilution, Microdilution…) Kirby-Bauer Method In accordance with standard and approved denominations	[79,80,84,85,86]
	Without forgetting to specify growing conditions (temperature and time of incubation, shaking or not)	
Bacteria	Specification of the origin of the bacteria: Strains issued from international collections (e.g., ATCC) or Clinical Isolates (with appropriated antibiograms) For example: *Escherichia coli* ATCC 25922 *Staphylococcus aureus* ATCC 29213 *Pseudomonas aeruginosa* ATCC 27853	
Bacterial medium	Mueller Hinton (MH)	
	Cation-Adjusted MH (CA-MH) Supplemented in accordance with the bacterial strains studied…	

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
