# Peer review of "Limitations of Recent Studies Dealing with the Antibacterial Properties of Silver Nanoparticles: Fact and Opinion"

_nanomaterials, 2019, doi:10.3390/nano9121775_

Round 1

Reviewer 1 Report

This review article summarizes the information from fifty-two publications since 2015 for antibacterial properties of AgNPs to propose recommendations for characterization and evaluation of AgNPs.

This paper would provide useful information to researchers who work on Nanomedicine. The reviewer thinks that the review will be improved if the following points are revised. 

I. As the authors pointed out, information should be treated differently between the characterization of AgNPs after preparation and the characterization of AgNPs at the study of antibacterial activity. Therefore, the reviewer wants to know the antibacterial data of AgNPs in each reference with conditions at the study because they would relate to the activity.

The table should contain the data of antibacterial activity in each reference with the final concentration of AgNPs in the culture media. The table containing the activity data is useful for researchers to know the size, shape, zeta-potential, or stabilizer effects on the antibacterial activity.  I agree with the authors claiming that the size and zeta-potential of AgNPs should be measured in the culture media. So it is useful to add the media used in the measurement to the table.   The information of aggregation and agglomeration of AgNPs under conditions in the study of antibacterial activity should also be added to the table.

II. The preparation method including stabilizers would characterize each work. So the reviewer would like to know in the text about the strategy to design AgNP formulation, i.e., to prevent aggregation, to control size, to control crystalline state, to modulate the surface accessibility, etc.

Cationic stabilizers such as CTAB or C12 Pyridinium would be used to provide positive zeta-potentials, but cationic surfactants themselves show some antibacterial activities.  Polymer stabilizers such as PVP, PVA, PEG would modulate the surface accessibility by their steric hindrance effects. 

III. About the evaluation of the antibacterial properties, the authors can refer the standards for in vitro evaluation of antimicrobial susceptibility from Clinical and Laboratory Standards Institute(CLSI). The points which should be discussed in the text would be different points of AgNPs from low-molecular weight antibacterial drugs.

The reviewer agree to use a broth microdilution method to calculate MIC for AgNPs. Is this method the same as a standard of CLSI? or need some modification for AgNPs? The reviewer is not familiar to use the cation-adjusted Mueller Hinton medium. Explain why this medium is recommendable. And is the CLSI standard also recommending this medium? If the broth microdilution method and CA-MH are noted in the standard, the description "There is currently no standard, no reference for studying the antibacterial activities of nanoparticles." is not accurate.

Author Response

Reviewer 1

This review article summarizes the information from fifty-two publications since 2015 for antibacterial properties of AgNPs to propose recommendations for characterization and evaluation of AgNPs.

This paper would provide useful information to researchers who work on Nanomedicine. The reviewer thinks that the review will be improved if the following points are revised.

As the authors pointed out, information should be treated differently between the characterization of AgNPs after preparation and the characterization of AgNPs at the study of antibacterial activity. Therefore, the reviewer wants to know the antibacterial data of AgNPs in each reference with conditions at the study because they would relate to the activity.

The table should contain the data of antibacterial activity in each reference with the final concentration of AgNPs in the culture media. The table containing the activity data is useful for researchers to know the size, shape, zeta-potential, or stabilizer effects on the antibacterial activity.

Author response:

Reviewer 1 is right; the details about the antibacterial activity are useful for researchers. However, as there are too many entries relating to antibacterial activities to add to Table 1, we decided to generate a new Table as a supporting material gathering all these data (i.e. Supp. Data 1).

I agree with the authors claiming that the size and zeta-potential of AgNPs should be measured in the culture media. So, it is useful to add the media used in the measurement to the table.

Author response:

We agree with reviewer 1 advise and added the media used in the measurement in Table 1.

The information of aggregation and agglomeration of AgNPs under conditions in the study of antibacterial activity should also be added to the table.

Author response:

We totally agree with reviewer 1, such information would be very useful. Unfortunately, we believe that we have not enough data at our disposal, this parameter is in fact very rarely noticed... Indeed, on the forty-nine selected papers, only two clearly inform the reader about the influence of the culture media on nanoparticles agglomeration/aggregation.

The preparation method including stabilizers would characterize each work. So, the reviewer would like to know in the text about the strategy to design AgNP formulation, i.e., to prevent aggregation, to control size, to control crystalline state, to modulate the surface accessibility, etc.

Author response:

We thank reviewer 1 for this comment. We think that the topic “strategy to design AgNP formulation“, which is a little bite out of our scope, deserves its own review. Indeed, the aim of our review is to highlight the aspects to improve, or at least to homogenize, in order to definitively establish the interest of silver nanoparticles as valuable antibacterial agents. So, we added recent references dealing with the design of NPs and biomedical applications, and modified the manuscript as follow:

“… and a stabilizer. Several recent review papers describe the different strategies to design NPs, more specifically, silver ones for biomedical applications [53–58]. In some cases, …“

Cationic stabilizers such as CTAB or C12 Pyridinium would be used to provide positive zeta-potentials, but cationic surfactants themselves show some antibacterial activities. Polymer stabilizers such as PVP, PVA, PEG would modulate the surface accessibility by their steric hindrance effects.

Author response:

Reviewer 1 is right; we agree with him and we thank him for his remark. In fact, we have removed the corresponding references from table 1.

III. About the evaluation of the antibacterial properties, the authors can refer the standards for in vitro evaluation of antimicrobial susceptibility from Clinical and Laboratory Standards Institute (CLSI). The points which should be discussed in the text would be different points of AgNPs from low-molecular weight antibacterial drugs.

The reviewer agrees to use a broth microdilution method to calculate MIC for AgNPs. Is this method the same as a standard of CLSI? or need some modification for AgNPs? The reviewer is not familiar to use the cation-adjusted Mueller Hinton medium. Explain why this medium is recommendable. And is the CLSI standard also recommending this medium? If the broth microdilution method and CA-MH are noted in the standard, the description "There is currently no standard, no reference for studying the antibacterial activities of nanoparticles." is not accurate.

Author response:

Clearly, to date there is no recommended method for evaluating the antibacterial properties of nanoparticles... none... For several years, within the laboratory, we have a team dedicated to the research and development of new anti-infectives and more particularly new antibacterials, and given the potential interest of nanoparticles, we decided to launch a research program on the subject. Unfortunately, as we have not found a reference method to evaluate the antibacterial activity of nanoparticles, we have transposed the method described for antibiotics, by CLSI, and we have even adapted it to specifically study the antibacterial activity nanoparticles (manuscript in preparation). So, it is for this reason that we have used the Cation-Adjusted Mueller Hinton medium because it is the medium that is described by the CLSI reference method. Therefore, we think the sentence: "There is currently no standard, no reference for studying the antibacterial activities of nanoparticles." is accurate.

Reviewer 2 Report

In this paper, the Authors review the methodologies used in recent years in the investigations of antibacterial activity of silver nanoparticles (AgNPs). This review is timely and will certainly help shaping the future research to focus on the proper design of the studies to achieve meaningful results and find solutions to unleash the power of AgNPs to fight the drug-resistant infections. A number of faults in otherwise very useful studies are listed and the improvements needed indicated, including the choice of bacteria used for testing AgNPs’ toxicity, size, shape and coatings of NPs, surface charge effects, influence of culture media on zeta potential, etc. The detailed analysis of research designs is followed by a comprehensive checklist suitable to guide future studies. This review is important for scientists working in microbiological area and nanoparticle toxicity.

I recommend the paper for publication after minor revision addressing the issues listed below.

Line 190: “Capping agent at the surface of the metallic core prevents NPs aggregation …”. The capping agents can either prevent aggregation or induce it, as shown for instance for glutathione (GSH)-capped AuNPs (M. Hepel and M. Stobiecka, 2012, Detection of Oxidative Stress Biomarkers Using Functional Gold Nanoparticles, in: Fine Particles in Medicine and Pharmacy, E. Matijevic [Ed.], Springer, Chapter 9, pp. 241-281, ISBN 978-1-4614-0378-4). Also, depending on pH of the medium and surface dissociation constants of the capping agents, the NPs’ assembly can occur (Sensors and Actuators B: Chemical, 149, 2010, pp. 373-380). Furthermore, the presence of divalent cations can induce assembly of NPs by bridging stable negatively charged colloidal particles (M. Hepel, D. Blake, M. McCabe, M. Stobiecka, K. Coopersmith, 2012, Assembly of Gold Nanoparticles Induced by Metal Ions, in: Functional Nanoparticles for Bioanalysis, Nanomedicine & Bioelectronic Devices, Volume 1, M. Hepel, C.J. Zhong, [Eds.], ACS Symposium Series Book, Oxford University Press, Inc, American Chemical Society, Washington, DC, Vol. 1112, Chapter 8, pp. 207-240; DOI: 10.1021/bk-2012-1112.ch008). These relevant literature references should be cited.

Typographical and English errors:

Line 251: “bacterium does not have of interest in human health” – please rephrase.

Author Response

Reviewer 2

In this paper, the Authors review the methodologies used in recent years in the investigations of antibacterial activity of silver nanoparticles (AgNPs). This review is timely and will certainly help shaping the future research to focus on the proper design of the studies to achieve meaningful results and find solutions to unleash the power of AgNPs to fight the drug-resistant infections. A number of faults in otherwise very useful studies are listed and the improvements needed indicated, including the choice of bacteria used for testing AgNPs’ toxicity, size, shape and coatings of NPs, surface charge effects, influence of culture media on zeta potential, etc. The detailed analysis of research designs is followed by a comprehensive checklist suitable to guide future studies. This review is important for scientists working in microbiological area and nanoparticle toxicity.

I recommend the paper for publication after minor revision addressing the issues listed below.

Line 190: “Capping agent at the surface of the metallic core prevents NPs aggregation …”. The capping agents can either prevent aggregation or induce it, as shown for instance for glutathione (GSH)-capped AuNPs (M. Hepel and M. Stobiecka, 2012, Detection of Oxidative Stress Biomarkers Using Functional Gold Nanoparticles, in: Fine Particles in Medicine and Pharmacy, E. Matijevic [Ed.], Springer, Chapter 9, pp. 241-281, ISBN 978-1-4614-0378-4). Also, depending on pH of the medium and surface dissociation constants of the capping agents, the NPs’ assembly can occur (Sensors and Actuators B: Chemical, 149, 2010, pp. 373-380). Furthermore, the presence of divalent cations can induce assembly of NPs by bridging stable negatively charged colloidal particles (M. Hepel, D. Blake, M. McCabe, M. Stobiecka, K. Coopersmith, 2012, Assembly of Gold Nanoparticles Induced by Metal Ions, in: Functional Nanoparticles for Bioanalysis, Nanomedicine & Bioelectronic Devices, Volume 1, M. Hepel, C.J. Zhong, [Eds.], ACS Symposium Series Book, Oxford University Press, Inc, American Chemical Society, Washington, DC, Vol. 1112, Chapter 8, pp. 207-240; DOI: 10.1021/bk-2012-1112.ch008). These relevant literature references should be cited.

Author response:

The references cited by reviewer 2 were added to the manuscript, and according to, the text was modified (Lines 194-197).

Typographical and English errors:

Line 251: “bacterium does not have of interest in human health” – please rephrase.

Author response:

Reviewer 2 is right. The sentence is rephrased as follow:

“Most of the time, these authors use this bacterium as a model, which may be conceivable, but then, it would be more scientifically right to be very careful about the interpretation of the results, e.g. by recalling that this bacterium does not have interest in human disease...”

and changed in the manuscript (Lines 256-259).

Reviewer 3 Report

Raphaël E. Duval, Jimmy Gouyau and Emmanuel Lamouroux reported a review manuscript entitled, “Limitations of recent studies dealing with the antibacterial properties of silver nanoparticles: a critical review” to Nanomaterials.

The authors extensively reviewed the characteristics of silver nanoparticles, and their antibacterial activities, how to evaluate antibacterial properties of silver nanoparticles and future perspectives.

First, why and how was the duration of the years 2015 to 2018 decided?

The style of this manuscript namely, “critical review”, seem to be very ambiguous and should be clarified. How were the contents in this manuscript selected and what is the rationale of the “selection” other than keywords?

Comparison of the properties and clinical benefits with excluded criteria such as other components close to this review’s silver nanoparticles such as biosynthesis/biogenic, composites (coated/coating, hybrid, alloys, core-shell, bi-metallic, decorated, combined to other metals such as titanium, copper or iron, grafted, filer, doped, modified, porous, hydrogel, etc.), carbonaceous materials (carbon nanotubes, graphene, etc.), nanoparticles, NPs, with a mean size higher than 100 nm (such as fibers, etc.), dental adhesive or supported/embedded nanoparticles (cellulose, silk, etc.) and all formulations using silver nanoparticles combined with antibacterial agent or active molecules (chitosan, pectin, tannic acid, curcumin, etc.), should be discussed.

Major drawbacks of silver nanoparticle including possible major complications possible died effects in specific applications should be elucidated and proposed how to prevent from it.

The longevities and minimal and maximal concentrations and doses should be presented.

Author Response

Reviewer 3

Raphaël E. Duval, Jimmy Gouyau and Emmanuel Lamouroux reported a review manuscript entitled, “Limitations of recent studies dealing with the antibacterial properties of silver nanoparticles: a critical review” to Nanomaterials.

The authors extensively reviewed the characteristics of silver nanoparticles, and their antibacterial activities, how to evaluate antibacterial properties of silver nanoparticles and future perspectives.

First, why and how was the duration of the years 2015 to 2018 decided?

Author response:

As we have written in the Introduction, research on nanoparticles, on their antibacterial properties, is a research subject in full expansion. Indeed, if we use the keywords "antibacterial", "silver" and "nanoparticles", we obtained more than 4700 articles, in English, via the SciFinder database... only for the period 2015-2018. Moreover, we found more than 100 reviews dealing with this research subject (i.e. antibacterial properties of silver nanoparticles) published before 2015. So, we had to make a choice, limiting ourselves to publications that are relatively recent.

The style of this manuscript namely, “critical review”, seem to be very ambiguous and should be clarified.

Author response:

Thank you to reviewer 3 for its comment. By choosing the title “Limitations of recent studies dealing with the antibacterial properties of silver nanoparticles: a critical review” for our review, we only want to underline that we have carefully selected and read more than 45 articles, in order to identify and discuss the limits of recent research studies dealing with the antibacterial properties of silver nanoparticles.

To be less ambiguous we can propose to reviewer 3 the following title: “Limitations of recent studies dealing with the antibacterial properties of silver nanoparticles: fact and opinion”.

How were the contents in this manuscript selected and what is the rationale of the “selection” other than keywords?

Comparison of the properties and clinical benefits with excluded criteria such as other components close to this review’s silver nanoparticles such as biosynthesis/biogenic, composites (coated/coating, hybrid, alloys, core-shell, bi-metallic, decorated, combined to other metals such as titanium, copper or iron, grafted, filer, doped, modified, porous, hydrogel, etc.), carbonaceous materials (carbon nanotubes, graphene, etc.), nanoparticles, NPs, with a mean size higher than 100 nm (such as fibers, etc.), dental adhesive or supported/embedded nanoparticles (cellulose, silk, etc.) and all formulations using silver nanoparticles combined with antibacterial agent or active molecules (chitosan, pectin, tannic acid, curcumin, etc.), should be discussed.

Author response:

Again, the publications dealing with the antibacterial activities of nanoparticles are very numerous. In addition, we focused on publications related to our own research theme: the synthesis, characterization and evaluation of the antibacterial properties of metallic nanoparticles obtained by chemical synthesis. That's why we have removed publications dealing with : “biosynthesis/biogenic (green, plants, extract, leaf, bacteria, fungi, etc.), composites (coated/coating, hybrid, alloys, core-shell, bi-metallic, decorated, combined to other metals such as titanium, copper or iron, grafted, filer, doped, modified, porous, hydrogel, etc.), carbonaceous materials (carbon nanotubes, graphene, etc.), NPs with a mean size higher than 100 nm (such as fibers, etc.), dental adhesive or supported/embedded nanoparticles (cellulose, silk, etc.) and all formulations using silver nanoparticles combined with antibacterial agent or active molecules (chitosan, pectin, tannic acid, curcumin, etc.)” (lines 54-60).

Major drawbacks of silver nanoparticle including possible major complications possible died effects in specific applications should be elucidated and proposed how to prevent from it.

Author response:

We agree with reviewer 3, several disadvantages for silver nanoparticles have been described in the literature. We can, in particular, think about the toxicity of nanoparticles, in general, and especially this of silver nanoparticles.

However, in this review, we wanted, above all, to confront several studies that are dealing with antibacterial properties of silver nanoparticles and see if it was possible to compare these studies with each other. In the end, it appears that this is not possible since these studies do not give us the same needed data about nanoparticles synthesis, nor on methods used for characterization, or methods used for the evaluation of the antibacterial properties; and in addition, when performed (i.e. antibacterial properties evaluation) bacteria are quite different (genus or species) and even, poorly characterized. So, the conclusion is simple: any comparison is impossible, and one can ask the question about whether or not silver nanoparticles have an antibacterial activity?

In the end, we believe that, before addressing the question of the toxicity of silver nanoparticles, it seems to us more important to have comparable studies between them, and to definitively rule on the potential interest of silver nanoparticles as available antibacterial agents.

Nevertheless, concerning the question of silver nanoparticles toxicity, one can recommend a very recent review on the subject published by MDPI :

Liao C, Li Y, Tjong SC. Bactericidal and Cytotoxic Properties of Silver Nanoparticles. Int J Mol Sci. 2019 Jan 21;20(2). pii: E449. doi:10.3390/ijms20020449.

The longevities and minimal and maximal concentrations and doses should be presented.

Author response:

We thank reviewer 3 for this comment. It is obvious that it will be necessary, at a moment, to ask the question of the evaluation of the silver nanoparticles in therapy, in humans. But we are still very far from this objective... Indeed, as long as we will not be able to have, at our disposal, perfectly characterized nanoparticles for which the antibacterial activity is precisely defined; it seems to us for the moment, unthinkable to use these compounds in humans.

Round 2

Reviewer 1 Report

The manuscript has been revised completely and the response to each comment is no problem.

Author Response

The manuscript has been revised completely and the response to each comment is no problem.

Author response:

We would like to thanks reviewer 1.

Reviewer 3 Report

Raphaël E. Duval, Jimmy Gouyau and Emmanuel Lamouroux reported a review manuscript entitled, “Limitations of recent studies dealing with the antibacterial properties of silver nanoparticles: fact and opinion” to Nanomaterials.

The authors reviewed the synthesis, characterization and evaluation of the antibacterial properties of metallic nanoparticles obtained by chemical synthesis.

Understanding focusing on specific themes and topics selected by the authors, extensive additions and re-writing are necessary.

The modes of actions in different species of the bacteria should be added.

The half-life properties and cellular metabolism analyses are necessary to be contained.

Overall discussion on “safety” issue to the “host” cells and tissue should be discussed.

Author Response

Reviewer 3

Understanding focusing on specific themes and topics selected by the authors, extensive additions and re-writing are necessary.

Author response:

Both reviewer 1 and reviewer 2 have accepted as it, our corrections; this comment is very surprising. Nevertheless, we have carefully reread our manuscript, and found no real problems… We are very grateful to reviewer 3 if he can give us examples.

The modes of actions in different species of the bacteria should be added

Author response:

As mentioned in the review, there are no proof at all of the exact mechanism of action of the silver nanoparticles that's why we have developed a specific paragraph on this item.

The half-life properties and cellular metabolism analyses are necessary to be contained.

Author response:

We are very sorry but we have a problem of comprehension, we do not understand what reviewer 3 asks…

Overall discussion on “safety” issue to the “host” cells and tissue should be discussed".

Author response:

Again, as mentioned in the first of corrections, the objectives of our review are to discuss about the processes of synthesis, characterization and the in vitro antibacterial properties of silver nanoparticles... we never aimed to discuss the toxicity of silver nanoparticles and we quoted a very recent review on this specific subject published by MDPI.

Nevertheless, in order to take into account the remark of reviewer we have modified the “Conclusions and futures perspective” section of our review and added a short paragraph on the feature of silver nanoparticles toxicity.
